# Conformational enantiodiscrimination for asymmetric construction of atropisomers

Shouyi Cen [1,2], Nini Huang[1,2], Dongsheng Lian[1], Ahui Shen [1], Mei-Xin Zhao [1] ✉ & Zhipeng Zhang [1] ✉

Molecular conformations induced by the rotation about single bonds play a crucial role in chemical transformations. Revealing the relationship between the conformations of chiral catalysts and the enantiodiscrimination is a formidable challenge due to the great difficulty in isolating the conformers. Herein, we report a chiral catalytic system composed of an achiral catalytically active unit and an axially chiral 1,1′-bi-2-naphthol (BINOL) unit which are connected via a C–O single bond. The two conformers of the catalyst induced by the rotation about the C–O bond, are determined via single-crystal X-ray diffraction and found to respectively lead to the formation of highly important axially chiral 1,1′-binaphthyl-2,2′-diamine (BINAM) and 2-amino-2′-hydroxy-1,1′-binaphthyl (NOBIN) derivatives in high yields (up to 98%), with excellent enantioselectivities (up to 98:2 e.r.) and opposite absolute configurations. The results highlight the importance of conformational dynamics of chiral catalysts in asymmetric catalysis.

Conformations are spatial arrangements of the atoms formed by rotations about a single bond. In most cases, pure conformers cannot be isolated, because the molecules are constantly rotating through all the possible conformations (Fig. 1a). When the rotation about a single bond is restricted, a special class of conformers called atropisomers can be isolated as different chemical species (Fig. 1b). Conformations and rotations about single bonds are crucial to molecular functions and chemical transformations. For example, conformational dynamics play a key role in enzyme catalysis[1,2]. However, in asymmetric catalysis, it is well-known that structural rigidity of non-enzymatic chiral catalysts, such as the 2,2′-bis(diphenylphosphino)−1,1′-binaphthyl (BINAP)-metal complexes[3,4] and BINOL-based phosphoric acids[5,6] shown in Fig. 1c, is usually essential in achieving high levels of asymmetric inductions[7–9] and less attention has been paid to the molecular flexibility and conformational dynamics of the chiral catalysts[10–15]. Due to the great difficulty in isolating the conformers, it is very challenging to reveal the relationship between the conformations of chiral catalysts and the enantiodiscrimination.

Axially chiral molecules not only are abundant in nature[16], but also make great success in many scientific fields such as materials science and asymmetric synthesis especially[17–24]. BINOL, BINAM, and NOBIN (Fig. 1b) are among the most prominent and valuable axially chiral molecules[17–24]. Enantiopure (R)- and (S)-BINOL and some of their derivatives are commercially available nowadays, in contrast, enantiopure BINAMs and NOBINs with diverse substitution patterns are still very difficult to obtain despite the fact that great efforts have been devoted to their synthesis in the past three decades[22–43].

In this work, we design a chiral catalytic system (Fig. 1d) which is composed of an achiral catalytically active unit (copper complex of 1,10-phenanthroline unit) and an axially chiral BINOL unit. The two units are connected via a C–O single bond, the rotation about which induces two distinct conformers. This dynamic catalytic system exhibits high activity and excellent enantioselectivity in the atroposelective synthesis of axially chiral BINAM and NOBIN derivatives which are highly important biaryl atropisomers. Moreover, the two conformers of the catalyst are determined via single-crystal X-ray diffraction and the relationship between the favored conformers and the

[1]Key Laboratory for Advanced Materials and Joint International Research Laboratory of Precision Chemistry and Molecular Engineering, Feringa Nobel Prize Scientist Joint Research Center, Frontiers Science Center for Materiobiology and Dynamic Chemistry, School of Chemistry and Molecular Engineering, East China University of Science & Technology, Shanghai 200237, China. [2]These authors contributed equally: Shouyi Cen, Nini Huang. ✉e-mail: mxzhao@ecust.edu.cn; zhipengzhang@ecust.edu.cn

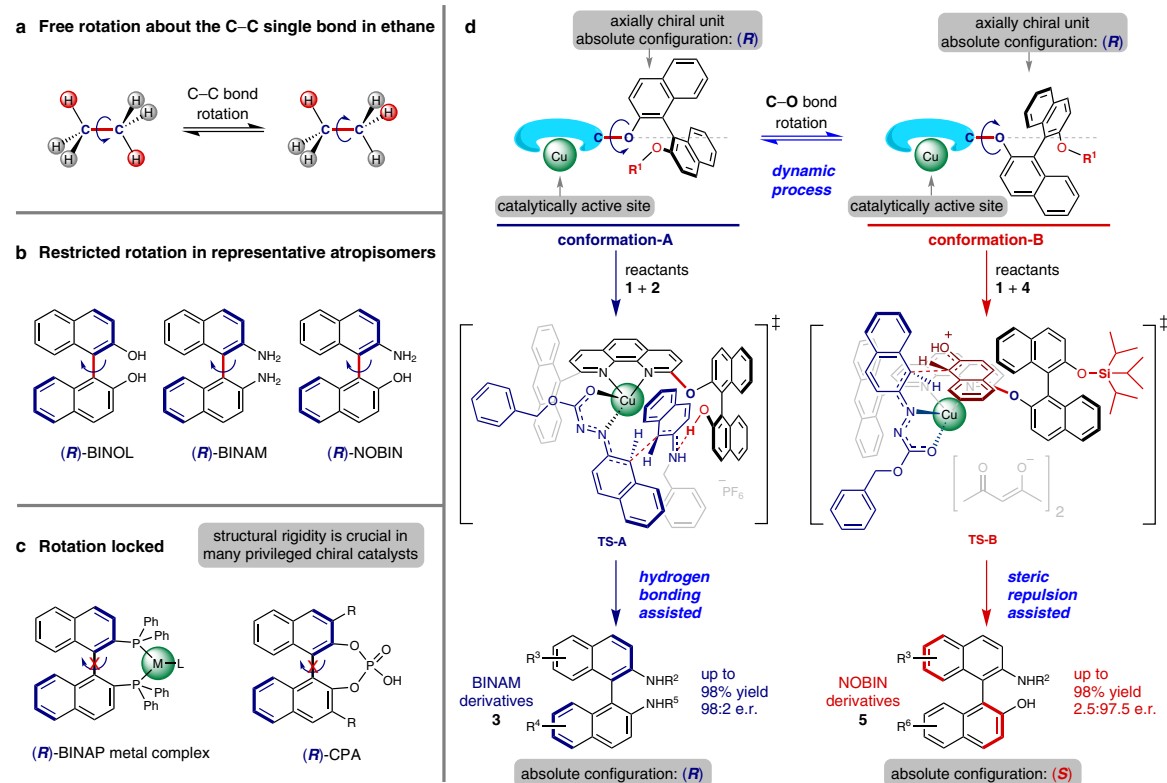

**Fig. 1 | Rotations about single bonds and conformational enantiodiscrimination. a** C–C bond rotation in ethane. **b** Restricted rotation about the C–C single bonds in atropisomers. **c** Rotation about the C–C single bond locked in BINAP-metal complexes and BINOL-based phosphoric acids. **d** Design of a chiral catalytic system for conformational enantiodiscrimination enabling asymmetric construction of atropisomers with opposite absolute configurations. BINOL, 1,1′-bi-2-naphthol; BINAM, 1,1′-binaphthyl-2,2′-diamine; NOBIN, 2-amino-2′-hydroxy-1,1′-binaphthyl; BINAP, 2,2′-bis(diphenylphosphino)-1,1′-binaphthyl; Ph, phenyl; CPA, chiral phosphoric acids.

enantiodiscrimination as well as the observed absolute configurations of the two classes of products is revealed.

## Results and discussion

### Asymmetric construction of BINAMs

To commence our investigation, we employed (*R*)-BINOL as the chiral unit and classical *N,N*-bidentate 1,10-phenanthroline as the achiral chelating unit. The two units were merged into a novel class of ligands via the formation of a C–O bond between the C2 carbon of the phenanthroline unit and the oxygen of one phenolic hydroxy group (see Section 2.1 in the Supplementary Information (SI) for details). Thus, the ligands are endowed with axial chirality, excellent coordination ability as well as conformational flexibility. In addition, the C9 position of the phenanthroline unit was left to be modified with sterically demanding groups which serve as a shield to narrow the chiral space around the metal center (Fig. 2a). The asymmetric cross-coupling of azonaphthalene **1a** with *N*-benzyl-2-naphthylamine **2a** which produces BINAM derivative **3a** was selected as the reaction[37] to evaluate the ligands (Fig. 2a) and copper, which has never been reported to catalyze this reaction, was employed as the metal.

We initiated our study by screening the substituents on the C9 position of the phenanthroline unit to find an appropriate shield (Fig. 2a). Although the reaction could hardly proceed under the optimized conditions (see Section 2.2 in the SI for optimization) when **L1** was employed as ligand, the desired product **3a** was obtained in 40% yield when **L2** which possesses a chlorine atom on the C9 position of the phenanthroline was used as ligand. However, the enantioselectivity is very low (52:48 e.r.). Introducing a phenyl substituent on the C9 position as the shield (**L3**) can dramatically improve the enantioselectivity (72:28 e.r.). Replacing the phenyl substituent with more sterically hindered 3,5-dimethylphenyl group (**L4**) did not improve the

enantioselectivity. Although replacing it with 3,5-di(trifluoromethyl) phenyl group (**L5**) improved the yield to 69%, the enantioselectivity is low. After evaluating the ligands (**L6-L10**) with fused aromatic ring on the C9 position of the phenanthroline, **L8** (9-anthracenyl as shield) was found to be the optimal ligand in terms of enantioselectivity, giving the desired product with 97.5:2.5 e.r. and in 67% yield. It is worthy to note that the absolute configuration of the major product is (*R*). Ligand **L11** with the same structure as **L8** except that the hydroxy group was converted to the methoxy group, was also tested under the same conditions (Fig. 2b). Surprisingly, both the yield (25%) and the enantioselectivity (47:53 e.r) decreased dramatically. These results indicate that possibly in the transition state hydrogen bond is formed between the substrate and the phenolic hydroxy group.

Further study revealed that extending the reaction time to 60 h increases the yield to 91% and the excellent e.r. (97.5:2.5) remains (Fig. 2c, product **3a**) utilizing **L8** as ligand. With the best ligand and optimized conditions in hand, we explored the scope of the produced BINAM derivatives (Fig. 2c). Replacing the benzyl moiety in the ester part of azonaphthalene with phenyl group renders **3b** in better yield but lower enantioselectivity. Whereas azonaphthalene with *n*-propyl ester group gives **3c** in lower yield with almost same level of enantioinduction (97:3). Azonaphthalenes with bromo, methyl, or ester group on C6 position and with bromo, methyl, phenyl, or methoxy group on C7 position are well tolerated. The corresponding BINAM derivatives **3d**–**3j** were obtained in yields ranging from 72 to 98% and with almost same level of enantioselectivities (from 96.5:3.5 to 97.5:2.5 e.r.). Introducing substituents on C6 or C7 position of 2-naphthylamines exhibits neglectable effect on the e.r. (ranging from 96:4 to 98:2) and products **3k**-**3q** were successfully obtained in moderate to excellent yields. 2-Naphthylamines with other similar protecting groups were also examined and all the corresponding products

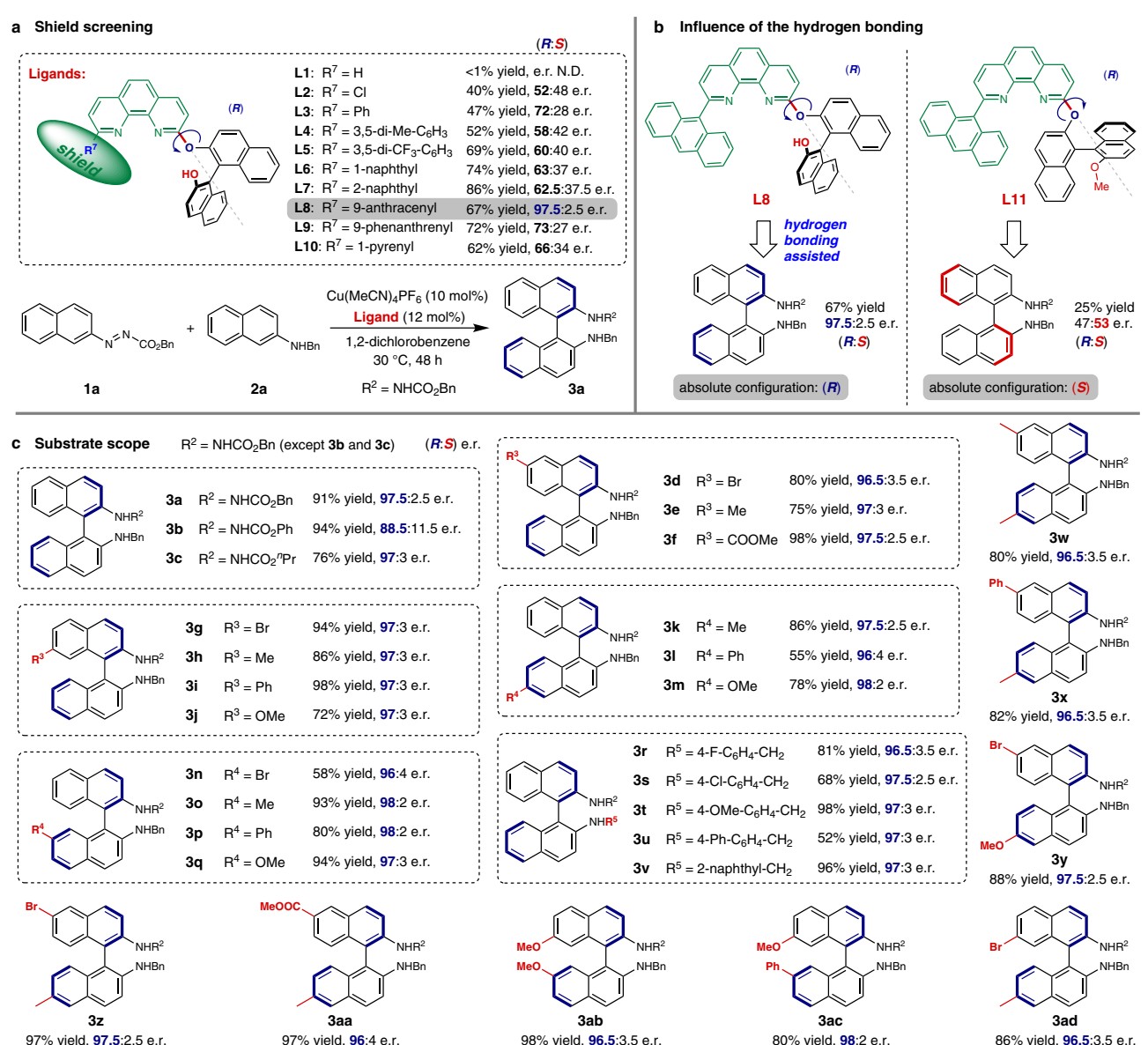

**Fig. 2 | Conformational enantiodiscrimination for the asymmetric cross-coupling of azonaphthalenes with 2-naphthylamines. a** Shield screening (48 h). **b** Influence of the hydroxy group (48 h). **c** Substrate scope (60 h). Reaction conditions: **1** (0.10 mmol), **2** (0.12 mmol), 1,2-dichlorobenzene (2.0 mL), under air, 30 °C unless noted otherwise (**3n**: 40 °C). All yields are isolated. Enantiomeric ratios (e.r.) were determined via HPLC analysis and reported as (*R:S*). Me, methyl; *n*Pr, *n*-propyl; Bn, benzyl; Ph, phenyl. The absolute configuration of all the products with blue thick bonds is *R*.

**3r**-**3v** were produced in excellent enantioselectivities (from 96.5:3.5 to 97.5:2.5 e.r.) and moderate to excellent yields. Furthermore, the catalyst is also compatible with various substituents in both azonaphthalenes and 2-naphthylamines simultaneously. Highly enantioenriched (from 96:4 to 98:2 e.r.) disubstituted BINAM derivatives **3w**-**3ad** have been successfully synthesized in good to excellent yields. To study the practicability of the protocol, a reaction was carried out on one-gram scale using **1a** (1.00 g, 3.45 mmol) and **2a** as reactants (Section 2.4 in the SI), producing **3a** in 79% yield and with excellent e.r. (97:3). In addition, product **3a** was successfully transformed to (*R*)-BINAM in 85% yield with the same e.r. (97:3) through Raney nickel-catalyzed hydrogenation under 1 atm (Section 2.5 in the SI). Thus, this catalytic system proves to be efficient for asymmetric synthesis of BINAM derivatives.

## Asymmetric construction of NOBINs

Encouraged by the previous results, we then targeted the asymmetric cross-coupling of azonaphthalene **1a** with 2-naphthol **4a** which produces NOBIN derivative **5a** (Fig. 3a)[37]. When the optimal ligand **L8** in the previous cross-coupling of azonaphthalenes with 2-naphthylamines was employed, in combination with Cu(acac)₂ to in-situ prepare the catalyst, good yield (70%) but poor enantioselectivity (66:34 e.r.) was obtained. The absolute configuration of the major product is determined to be (*R*), which is the same as those of products **3** in cross-coupling of azonaphthalene **1a** with *N*-benzyl-2-naphthyla-mine **2a** (Fig. 2). Since 2-naphthol is an acidic substrate, the hydrogen bonding may be interrupted which may partly account for the observed low enantioinduction. Both the yield (78%) and the enantioselectivity (68.5:31.5) are slightly improved in the presence of 20 mol% NaHCO₃, possibly due to an enhanced hydrogen bonding (see Supplementary Table 8). Interestingly, ligand **L11** gives better results (89% yield and 24.5:75.5 e.r.) compared with **L8**. More importantly, the absolute configuration of the major product is (*S*). We speculate that in the cross-coupling of azonaphthalenes with 2-naphthols, the azonaphthalene is activated by the copper center and could react with 2-naphthol directly. Hydrogen bonding interaction is not

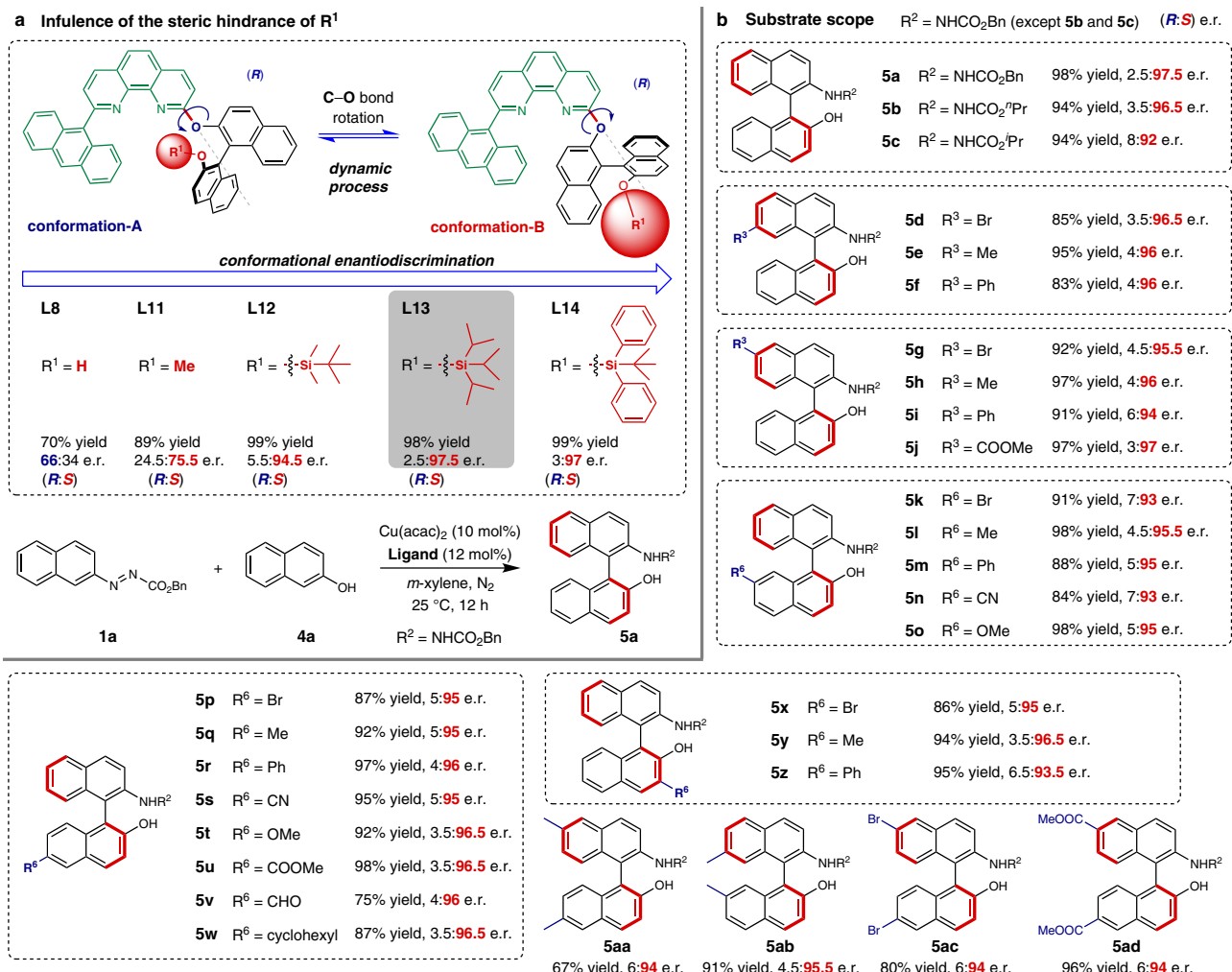

**Fig. 3 | Conformational enantiodiscrimination for the asymmetric cross-coupling of azonaphthalenes with 2-naphthols. a** Influence of the conformational equilibrium on enantiodiscrimination. **b** Substrate scope. Reaction conditions: **1** (0.10 mmol), **4** (0.12 mmol), Cu(acac)₂ (10 mol%), ligand (12 mol%), *m*-xylene (2.0 mL), under N₂, at 25 °C. All yields are isolated. Enantiomeric ratios (e.r.) were determined via HPLC analysis and reported as (*R*:*S*). Me, methyl; *ⁿ*Pr, *n*-propyl; *ⁱ*Pr, *iso*-propyl; Bn, benzyl; Ph, phenyl. The absolute configuration of all the products with red thick bonds is *S*.

indispensable in the transition state, and the steric repulsion generated from the methoxy group may shift the conformational equilibrium of the ligand and favor the distribution of conformation-B. Therefore, to enhance the dominance of conformation-B so as to obtain more efficient enantiodiscrimination, more sterically hindered *tert*-butyldimethylsilyl, triisopropylsilyl, and *tert*-butyldiphenylsilyl groups were employed to protect the phenolic hydroxy group, and ligands **L12**–**L14** were synthesized and evaluated (Fig. 3a). All these three ligands display excellent yields and enantioselectivities (e.r. up to 2.5:97.5 for **L13**). These results are consistent with the designed conformation-controlled enantiodiscrimination model (Fig. 1d) and the observed absolute configuration of the major product.

This catalytic system (**L13**/Cu(acac)₂) was further evaluated with various substituted azonaphthalenes and 2-naphthols (Fig. 3b). In most of the cases, the transformations proceeded smoothly and the corresponding NOBIN derivatives were obtained in excellent yields and enantioselectivities under the optimized conditions (see Section 3.2 in the SI for optimization). Replacing the benzyl moiety in the ester part of azonaphthalene with *n*-propyl or *iso*-propyl group provides the products **5b** and **5c** respectively, in slightly reduced yields and enantioselectivities. Azonaphthalenes with bromo, methyl or phenyl at either C7 or C6 position of the naphthalene ring are compatible, affording the corresponding products (**5d**-**5i**) in good to excellent

yields (83–97%) and excellent e.r. (up to 3.5:96.5). Introducing a methyl ester group on the C6 position gave the product (**5j**) in 97% yield and with an e.r. of 3:97. In addition, the effect of substituents on the 2-naphthol were also examined. 2-Naphthols with bromo, methyl, phenyl, cyano or methoxy group on either C7 or C6 position were tolerated well, and products **5k**-**5t** were produced in good to excellent yields (84–98%) and enantioselectivities (up to 3.5:96.5 e.r.). Methyl ester, formyl and cyclohexyl groups on C6 position of 2-naphthol are also compatible, affording product **5u** in 98%, **5v** in 75%, and **5w** in 87% yield respectively with almost identical level of enantioinductions. More importantly, substituents (bromo, methyl, especially phenyl) on C3 position of 2-naphthol were well-tolerated, rendering the products (**5x**-**5z**) in good to excellent yields and enantioselectivities. Substrates with methyl, bromo, or methyl ester group on both azonaphthalenes and 2-naphthol were tested and corresponding NOBIN derivatives (**5aa**-**5ad**) were successfully synthesized with good to excellent enantioinductions. A gram-scale reaction for asymmetric cross-coupling of **1a** (1.16 g, 4.00 mmol) and **4a** was also carried out (Section 3.4 in the SI), producing **5a** in 96% yield and with excellent e.r. (6:94). Furthermore, product **5a** can be transformed to (*S*)-NOBIN in 96% yield and with 6:94 e.r. through Raney nickel-catalyzed hydrogenation (Section 3.5 in the SI). Therefore, the catalytic system can also be applied to the atroposelective construction of NOBIN derivatives.

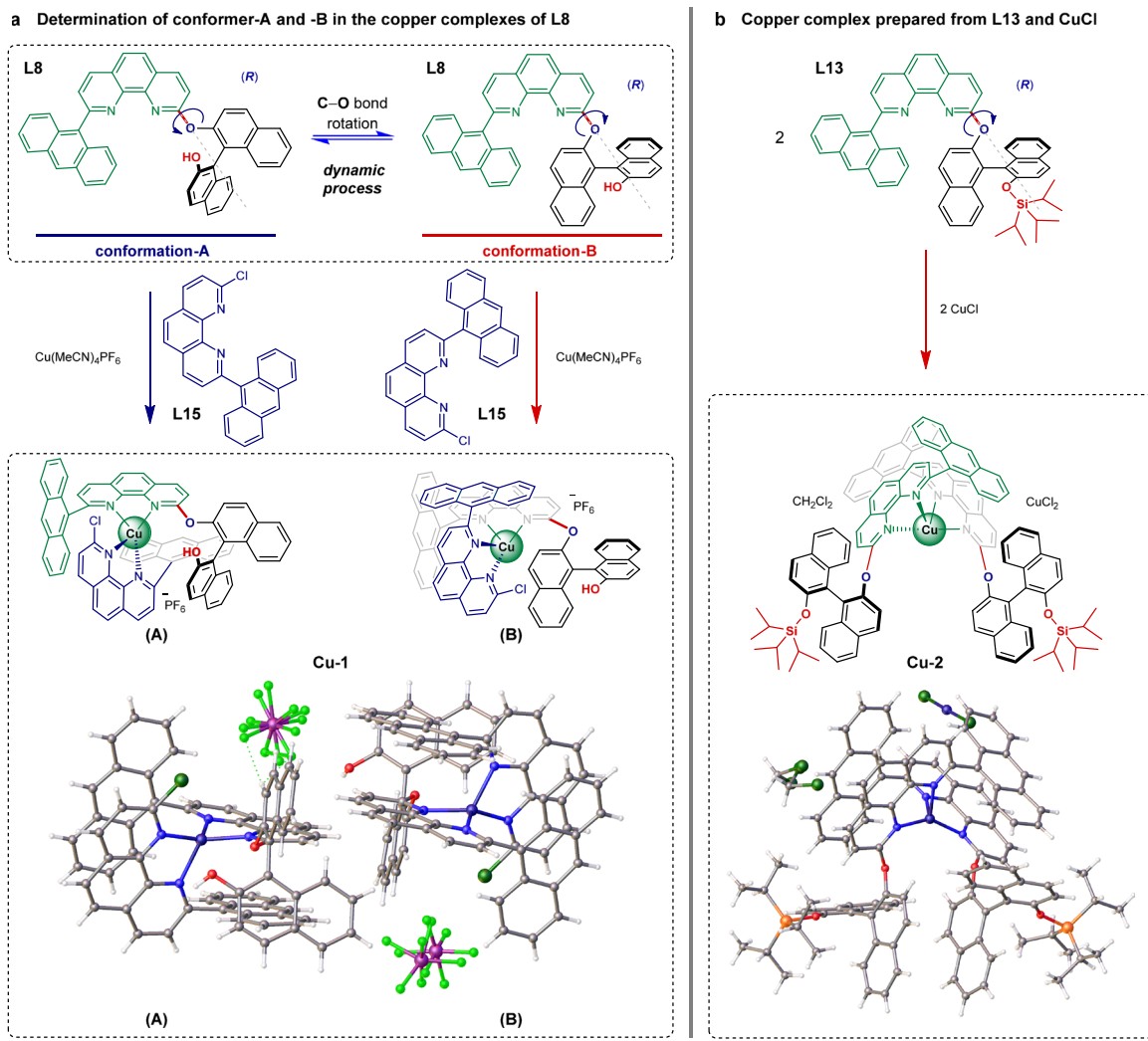

**Fig. 4 | Determination of the conformers via single-crystal X-ray diffraction.**
**a** Copper complex **Cu-1** prepared from **L8**, Cu(MeCN)$_4$PF$_6$ and 2-(anthracen-9-yl)

−9-chloro-1,10-phenanthroline (**L15**). **b** Copper complex **Cu-2** prepared from **L13** and CuCl.

## Determination of the conformers

To determine the major conformations of this chiral catalytic system induced by the rotation about the C−O bond, single-crystal X-ray diffraction was employed to determine the structure of copper complexes prepared from the optimal ligands. A single crystal (**Cu-1**) was successfully obtained by using 2-(anthracen-9-yl)−9-chloro-1,10-phenanthroline (**L15**) to stabilize the catalyst prepared from **L8** and Cu(MeCN)$_4$PF$_6$. As shown in Fig. 4a, two distinct conformers of **L8** induced by the rotation of the carbon-oxygen single bond which connects the BINOL unit with the phenanthroline unit, were observed in the crystal structures (corresponding to two copper complexes co-crystallized in 1:1 ratio). In **Cu-1**(A) the hydroxy group points towards the front (conformation-A) and the anthracene group of the stabilizer **L15** is under the phenanthroline unit. In **Cu-1**(B), the hydroxy group extends towards the back (conformation-B) and the anthracene group of **L15** is above the phenanthroline unit. This indicates the conformations of the ligand could discriminate the coordination modes of the substrate with the catalyst if we imagine that substrate **1a** would take the place of **L15** during the catalytic process. Since previous results proved the hydroxy group is crucial to the reactivity and enantioselectivity, conformation-A shown in **Cu-1**(A) is believed to be favorable and productive. By reacting **L13** with CuCl (1.0 equiv.) in methanol and recrystallizing from dichloromethane/*n*-hexane, a copper complex composed of Cu(**L13**)$_2$ and CuCl$_2$ was obtained. The crystal structure

of **Cu-2** in Fig. 4b shows two **L13** coordinates to one copper in a criss-cross pattern, forming a $C_2$-symmetric complex in which the bulky triisopropylsilyl groups point away from the metal center (similar to the conformation-B shown in **Cu-1**(B)). Conformation-A was not observed in the solid state of **L13**, and conformation-B is proposed to be the favored one during catalysis. A shift in the conformational equilibrium of the ligands may occur when the steric hindrance of R$^1$ increases (Figs. 1d and 3a). These crystallographic data are in agreement with the proposed possible transition states (**TS-A** and **TS-B**) shown in Fig. 1d and the observed conformation-controlled enantiodiscrimination in the two reactions for the asymmetric construction of BINAM and NOBIN derivatives (Figs. 2 and 3).

In summary, conformational flexibility has been incorporated into the design and development of a chiral catalytic system which proves to be efficient and highly enantioselective for the atroposelective synthesis of highly valuable axially chiral BINAM and NOBIN derivatives. The relationship between the conformational preference of the catalysts and the conformation-controlled enantiodiscrimination has been revealed. The absolute configuration of the products is determined by the conformation of the catalysts rather than the absolute configuration of the BINOL unit. The findings in this study highlight the importance of conformational dynamics of chiral catalysts in asymmetric catalysis and may inspire future development of other chiral catalysts.

## Methods

### General procedure for the asymmetric synthesis of BINAM derivatives

To a solution of Cu(MeCN)$_4$PF$_6$ (3.7 mg, 0.010 mmol, 10 mol%) and **L8** (7.7 mg, 0.012 mmol, 12 mol%) in 1,2-dichlorobenzene (2.0 mL) were added azo compound **1** (0.10 mmol) and the 2-naphthylamine derivative **2** (0.12 mmol). The mixture was stirred under air at 30 °C for 60 h. Upon completion, the resulting mixture was directly purified by flash chromatography on silica gel using petroleum ether/ethyl acetate as the eluent to afford the desired products **3**.

### General procedure for the asymmetric synthesis of NOBIN derivatives

To a solution of Cu(acac)$_2$ (2.6 mg, 0.010 mmol, 10 mol%) and **L13** (9.6 mg, 0.012 mmol, 12 mol%) in *m*-xylene (2.0 mL) were added azo compound **1** (0.10 mmol) and the 2-naphthylamine derivative **4** (0.12 mmol). The mixture was stirred under N$_2$ atmosphere at 25 °C for 12 h. Upon completion, the resulting mixture was directly purified by flash chromatography on silica gel using petroleum ether/ethyl acetate as the eluent to afford products **5**.

## Data availability

The data supporting the findings of this study are available within the paper and its Supplementary Information. Metrical parameters for the structure of copper complexes (**Cu-1** and **Cu-2** in Fig. 4) (see Supplementary Information) are available free of charge from the Cambridge Crystallographic Data Centre (https://www.ccdc.cam.ac.uk/) under reference numbers CCDC 2096699 and CCDC 2096715, respectively. Any further relevant data are available from the authors on request.

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

## Acknowledgements
We thank the Research Center of Analysis and Test, East China University of Science and Technology for the help on the characterization, and Prof. Yifeng Chen (ECUST) for sharing the instruments. We thank Prof. Yuan-Yuan Zhu (HFUT) and Prof. Jie Sun (SIOC) for the help with single crystal X-ray diffraction and structure analysis. We acknowledge the National Natural Science Foundation of China (21702059), Fundamental Research Funds for the Central Universities (222201814014, JKVJ1211010, JKVJ12001010), Shanghai Pujiang Program (18PJ1402200), Shanghai Municipal Science and Technology Major Project (2018SHZDZX03), Program of Introducing Talents of Discipline to Universities (B16017) and the "Thousand Plan" Youth program for financial support.

## Author contributions
Z.Z. conceived of and directed the project. S.C. developed the BINAM part, N.H. developed the NOBIN part, S.C. and D.L. performed the gram-scale reactions. S.C., N.H., A.S., M.X.Z. and Z.Z. cowrote the manuscript.

## Competing interests
The authors declare no competing interests.
