## [Peer Review File · Nature Communications]

REVIEWER COMMENTS

Reviewer #1 (Remarks to the Author):

This manuscript describes the atroposelective synthesis of BINAM and NOBIN derivatives via copper/chiral ligand complex-catalyzed addition of azonaphthalenes with 2-naphthylamines or 2-naphthols as nucleophiles. Both reactions feature mild conditions, broad substrate scope as well as generally excellent enantiocontrol. Additionally, the delivered biaryl products are of significant importance in various research fields. As we know, catalytic asymmetric synthesis of BINAM and NOBIN derivatives has never been an easy task in synthetic chemistry. At first glance, the work presented here is quite similar to that was reported by the Tan group (Nat. Catal. 2019, 2, 314; also see ref. 36). However, the catalytic system and asymmetric induction mode are entirely different. Notably, both types of the nucleophiles were well accommodated through slight modification of the substituent on BINOL-based axially chiral ligand. When 2-naphthylamines were utilized the free hydroxyl group on BINOL-core structure is necessary to acquire satisfied enantioselectivity, while OH should be masked with sterically hindered group for enhancing the enantiocontrol with 2-naphthols. The authors also got the X-ray structure of the catalysts to illustrate the relationship between conformation and enantiodiscrimination. Overall, this work provides an effective and facile alternative to approach the highly valuable BINAM and NOBIN derivatives, and the manuscript is well prepared. I would like to give my strong support on the publication of this work on Nature Communications with minor revision.

(1) Figure 1d and Figure 4a, blue birds were inserted in the chemical structures. This reviewer suggests that it is better to remove them if there is no related explanation or description.

(2) Line 106-111, the authors stated that 'Hydrogen bonding interaction is not indispensable in the transition state' to explain 'why ligand L11 gives better results compared with L8?'. H-bond between HN and HO of the BINOL was proposed in the stereocontrol mode for the reaction with 2-naphthylamines. However, when 2-naphthol substrates were utilized as the nucleophiles, the acidity of 2-naphthol is quite similar to that of BINOL. In this case, the H-bond may be interrupted. This point should be taken into consideration here.

(3) Linked to the above, did the authors try the reaction of azonaphthalene and 2-naphthol with addition of catalytic amount of weak base?

(4) For the gram-scale reaction of 1a and 4a, 5a was obtained with an e.r. of 6:94, while the e.r. was 2.5:97.5 in Figure 3b. The deterioration of the enantiocontrol is distinct. Did the authors repeat this reaction?

Reviewer #2 (Remarks to the Author):

This manuscript describes the development of ligands for the reactions that were reported in reference 36. A phenanthroline was coupled to BINOL by a C-O bond and different shielding groups were installed. BINAMs were prepared after optimizing the ligand and an effect of the hydroxy group was observed (OH compared to OMe). Reversal of selectivity was found with bulky groups and NOBINs were prepared again with the strategy reported in reference 36. The effect of the OH and OSiR3 groups were further investigated by X-ray crystallography.

The paper is overall interesting, but the observed effect was not surprising. The title and abstract initially suggested a conceptual advance, but the results are more of a technical nature. It is therefore suggested to submit this paper to a more specialized journal.

Reviewer #3 (Remarks to the Author):

The authors describe the method for the synthesis of BINAM (1,1'-binaphthyl-2,2'-diamine) and NOBIN (2-amino-2'-hydroxy-1,1'-binaphthyl) derivatives using a newly designed chiral catalytic system, which is composed of an achiral catalytically active unit and an axially chiral BINOL unit via a C-O single bond connection.

This manuscript has several merits as described below: (i) Chiral BINOL-tethered phenanthroline plays vital roles on creating chiral environment by the complexations of copper ion with phenanthroline and benzyl diazocarbonylate. For the synthesis of BINAMs, H-bonding between free hydroxy group in the chiral BINOL and the nitrogen of 2-naphthylamine in the presence of Cu-complexes are essential to achieve high enantioselection. For the synthesis of NOBINs, the steric repulsion, which is induced by the introduction of bulkier silyl group on the hydroxy group of BINOL, is highly demanded, leading to the construction of conformation-controlled enantiodiscrimination model. Such catalytic system provides to be efficient for the synthesis of atropisomers and has not been reported in the literature so far! (ii) Their approach is also systematically conducted with the full investigation of parameters and reaction scopes. In addition, analytical data including ¹H and ¹³C NMR spectroscopy, HRMS, and HPLC spectra are well organized.

The reviewer recommends this manuscript for publication in the Nature Communications because the novelty of this work meets the standards of this journal. The following comments should be addressed before acceptance of the article.

1. To help a better understanding of the protocol for readers, the synthetic routes for the conversion of 3a to BINAM and the conversion of 5a to NOBIN should be briefly mentioned within the manuscript somewhere.

2. The following key reference on the application of privileged chiral BINOL-derived organocatalysts should be included in the Reference Section: List, B. & Yang, J. W. The organic approach to asymmetric catalysis. *Science* 313, 1584-1586 (2006).

REVIEWER COMMENTS AND AUTHOR RESPONSES

Reviewer #1 (Remarks to the Author):

This manuscript describes the atroposelective synthesis of BINAM and NOBIN derivatives via copper/chiral ligand complex-catalyzed addition of azonaphthalenes with 2-naphthylamines or 2-naphthols as nucleophiles. Both reactions feature mild conditions, broad substrate scope as well as generally excellent enantiocontrol. Additionally, the delivered biaryl products are of significant importance in various research fields. As we known, catalytic asymmetric synthesis of BINAM and NOBIN derivatives has never been an easy task in synthetic chemistry. At first glance, the work presented here is quite similar to that was reported by the Tan group (Nat. Catal. 2019, 2, 314; also see ref. 36). However, the catalytic system and asymmetric induction mode are entirely different. Notably, both types of the nucleophiles were well accommodated through slight modification of the substituent on BINOL-based axially chiral ligand. When 2-naphthylamines were utilized the free hydroxyl group on BINOL-core structure is necessary to acquire satisfied enantioselectivity, while OH should be masked with sterically hindered group for enhancing the enantiocontrol with 2-naphthols. The authors also got the X-ray structure of the catalysts to illustrate the relationship between conformation and enantiodiscrimination. Overall, this work provides an effective and facile alternative to approach the highly valuable BINAM and NOBIN derivatives, and the manuscript is well prepared. I would like to give my strong support on the publication of this work on Nature Communications with minor revision.

Response: We thank this reviewer very much for the very positive comments. We appreciate this reviewer's strong support on the publication of this work in Nature Communications.

(1) Figure 1d and Figure 4a, blue birds were inserted in the chemical structures. This reviewer suggests that it is better to remove them if there is no related explanation or description.

Response: We thank this reviewer for the kind suggestion. The blue birds have been removed from the two figures as suggested.

(2) Line 106-111, the authors stated that 'Hydrogen bonding interaction is not indispensable in the transition state' to explain 'why ligand L11 gives better results compared with L8?'. H-bond between HN and HO of the BINOL was proposed in the stereocontrol mode for the reaction with 2-naphthylamines. However, when 2-naphthol substrates were utilized as the nucleophiles, the acidity of 2-naphthol is quite similar to that of BINOL. In this case, the H-bond may be interrupted. This point should be taken into consideration here. (3) Linked to the above, did the authors try the reaction of azonaphthalene and 2-naphthol with addition of catalytic amount of weak base?

Response: We thank this reviewer for pointing this out and we quite agree that the acidity of 2-naphthol may interrupt the hydrogen bonding. This point has been taken into consideration and we have conducted several experiments with azonaphthalene **1a** and 2-naphthol **4a** as substrates, **L8** as ligand, in the presence of catalytic amount of base. As shown in the following table, both the yield (78%) and the enantioselectivity (68.5:31.5) are slightly improved in the presence of 20 mol% NaHCO₃, possibly due to an enhanced hydrogen bonding. Since ligand **L11** (without a phenolic hydroxy group) gives even higher yield (89%) and better enantioselectivity (24.5:75.5 e.r.), the reaction seems to be able to proceed smoothly without a hydrogen bonding.

The follow sentence has been added to the main text. "Since 2-naphthol is an acidic substrate, the hydrogen bonding may be interrupted which may partly account for the observed low enantioinduction. Both the yield (78%) and the enantioselectivity (68.5:31.5) are slightly improved in the presence of 20 mol% NaHCO₃, possibly due to an enhanced hydrogen bonding (see Supplementary Table 8)."

Supplementary Table 8. The effect of a base on the reaction using L8 as ligand^a

Entry	Base (20 mol%)	Yield ^b (%)	E.r. ^c (R : S)
1	/	70	66:34
2	NaHCO ₃	78	68.5:31.5
3	Na ₂ CO ₃	60	66:34
4	Cs ₂ CO ₃	69	58.5:41.5
5	DMAP	78	67:33

^aAll reactions were carried out on 0.1 mmol scale in *m*-xylene (2.0 mL) at 25 °C for 12 h. ^bYields of the isolated products. ^cEnantiomeric ratios (e.r.) were determined via HPLC on a chiral stationary phase and reported as (*R*:*S*).

(4) For the gram-scale reaction of **1a** and **4a**, **5a** was obtained with an e.r. of 6:94, while the e.r. was 2.5:97.5 in Figure 3b. The deterioration of the enantiocontrol is distinct. Did the authors repeat this reaction?

Response: Actually, besides the gram-scale reaction of **1a** and **4a**, we had also conducted a 0.28-gram-scale reaction and **5a** was obtained in 95% yield and with a slightly deteriorated e.r. (4.5:95.5). We also repeated the gram-scale reaction and similar results (95% yield, 6:94 e.r.) were obtained.

We thank this reviewer for the kind suggestions and for helping improve our manuscript.

Reviewer #2 (Remarks to the Author):

This manuscript describes the development of ligands for the reactions that were reported in reference 36. A phenanthroline was coupled to BINOL by a C-O bond and different shielding groups were installed. BINAMs were prepared after optimizing the ligand and an effect of the hydroxy group was observed (OH compared to OMe). Reversal of selectivity was found with bulky groups and NOBINs were prepared again with the strategy reported in reference 36. The effect of the OH and OSiR3 groups were further investigated by X-ray crystallography. The paper is overall interesting, but the observed effect was not surprising. The title and abstract initially suggested a conceptual advance, but the results are more of a technical nature. It is therefore suggested to submit this paper to a more specialized journal.

Response: We thank this reviewer for concluding that this manuscript is interesting. We respectfully disagree with the reviewer's comments on the title, abstract and the results. As we known, catalytic asymmetric synthesis of BINAM and NOBIN derivatives has never been an easy task in synthetic chemistry. The catalytic system and asymmetric induction mode in our study are entirely different from those in Tan's report. Both types of the nucleophiles (2-naphthylamines and 2-naphthols) were well accommodated with one novel catalytic system. More importantly, our results (Figure 2b, 3a, 4a) clearly show that the absolute configuration of the products is determined by the conformation of the catalysts rather than the absolute configuration of the BINOL unit. The relationship between the conformational preference of the catalysts and the conformation-controlled enantiodiscrimination has been clearly revealed. The findings in this study highlight the importance of the conformational dynamics of chiral catalysts in asymmetric catalysis, and are in good agreement with the title and abstract. We believe this manuscript is suitable for publishing in Nature Communications.

Reviewer #3 (Remarks to the Author):

The authors describe the method for the synthesis of BINAM (1,1'-binaphthyl-2,2'-diamine) and NOBIN (2-amino-2'-hydroxy-1,1'-binaphthyl) derivatives using a newly designed chiral catalytic system, which is composed of an achiral catalytically active unit and an axially chiral BINOL unit via a C-O single bond connection.

This manuscript has several merits as described below: (i) Chiral BINOL-tethered phenanthroline plays vital roles on creating chiral environment by the complexations of copper ion with phenanthroline and benzyl diazocarbonylate. For the synthesis of BINAMs, H-bonding between free hydroxy group in the chiral BINOL and the nitrogen of 2-naphthylamine in the presence of Cu-complexes are essential to achieve high enantioselection. For the synthesis of NOBINs, the steric repulsion, which is induced by the introduction of bulkier silyl group on the hydroxy group of BINOL, is highly demanded, leading to the construction of conformation-controlled enantiodiscrimination model. Such catalytic system provides to be efficient for the synthesis of atropisomers and has not been reported in the literature so far! (ii) Their approach is also systematically conducted with the full investigation of parameters and reaction scopes. In addition, analytical data including ¹H and ¹³C NMR spectroscopy, HRMS, and HPLC spectra are well organized.

The reviewer recommends this manuscript for publication in the Nature Communications because the novelty of this work meets the standards of this journal. The following comments should be addressed before acceptance of the article.

Response: We thank this reviewer very much for recognizing the novelty and quality of this work. We appreciate this reviewer's support on publication of this work in Nature Communications.

1. To help a better understanding of the protocol for readers, the synthetic routes for the conversion of 3a to BINAM and the conversion of 5a to NOBIN should be briefly mentioned within the manuscript somewhere.

Response: We thank this reviewer for the kind suggestions. The conversion of **3a** to BINAM and **5a** to NOBIN, have been briefly mentioned in the corresponding part of the manuscript.

“In addition, product **3a** was successfully transformed to (*R*)-BINAM in 85% yield with the same e.r. (97:3) through Raney nickel-catalyzed hydrogenation under 1 atm (Section 2.5 in the SI).”

“Furthermore, product **5a** can be transformed to (*S*)-NOBIN in 96% yield and with 6:94 e.r. through Raney nickel-catalyzed hydrogenation (Section 3.5 in the SI).”

2. The following key reference on the application of privileged chiral BINOL-derived organocatalysts should be included in the Reference Section: List, B. & Yang, J. W. The organic approach to asymmetric catalysis. Science 313, 1584-1586 (2006).

Response: We thank this reviewer for the kind suggestion. The mentioned reference has been included as ref-7 in the manuscript.

We thank this reviewer for the very positive comments and for helping improve our manuscript.

REVIEWERS' COMMENTS

Reviewer #1 (Remarks to the Author):

The authors have addressed most of points raised. This reviewer would like to recommend the publication of this work in Nature Communications without any delay.